# Impact of Multidrug-Resistant Organisms on Severe Acquired Brain Injury Rehabilitation: An Observational Study

**DOI:** 10.3390/microorganisms12040830

**Published:** 2024-04-19

**Authors:** Giovanna Barbara Castellani, Elisa Maietti, Valentina Colombo, Stefano Clemente, Ivo Cassani, Paola Rucci

**Affiliations:** 1Montecatone Rehabilitation Institute, 40026 Imola, Italy; valentina.colombo@montecatone.com (V.C.); stefano.clemente@montecatone.com (S.C.); ivo.cassani@montecatone.com (I.C.); 2Department of Biomedical and Neuromotor Sciences, University of Bologna, 40126 Bologna, Italy; paola.rucci2@unibo.it

**Keywords:** multidrug resistance, carbapenemase-producing Enterobacteriaceae, infections, rehabilitation outcome, brain injuries

## Abstract

Healthcare-associated infections (HAIa) and antimicrobial resistance are expected to be the next threat to human health and are most frequent in people with severe acquired brain injury (SABI), who can be more easily colonized by multidrug-resistant organisms (MDROs). The study’s aim is to investigate the impact of MDRO colonizations and infections on SABI rehabilitation outcomes. This retrospective observational study was performed in a tertiary referral specialized rehabilitation hospital. The main outcomes were the presence of carbapenemase-producing Enterobacteriaceae (CPE) colonization, type and timing of HAI and MDRO HAI, and the number of CPE transmissions. We included 48 patients, 31% carrying CPE on admission and 33% colonized during the hospitalization. A total of 101 HAI were identified in 40 patients, with an overall incidence of 10.5/1000 patient days. Some 37% of patients had at least one MDRO infection, with a MDRO infection incidence of 2.8/1000 patient days. The number of HAIs was significantly correlated with the length of stay (LOS) (r = 0.453, *p* = 0.001). A significant correlation was found between colonization and type of hospital room (*p* = 0.013). Complications and HAI significantly affected LOS. We suggest that CPE carriers might be at risk of HAI and worse outcomes compared with non-CPE carriers.

## 1. Introduction

The COVID-19 pandemic has challenged healthcare delivery worldwide, highlighting the shortcomings of some national healthcare systems in their ability to meet the needs the frail [1]. Similarly, antimicrobial resistance is expected to be the next threat to human health and economies, with levels in countries in the Organization for Economic Co-operation and Development expected to be 70% higher in 2030 than in 2005. In Europe, North America and Australia, antimicrobial resistance could kill around 2.4 million people between 2015 and 2050 unless rapid and effective action is taken [2].

One of the most urgent public health needs identified by the World Health Organization in its “Global priority list of antibiotic-resistant bacteria to guide research, discovery, and development of new antibiotics” is the treatment of multidrug-resistant bacteria. They pose a particular threat in hospitals, in nursing homes, and to patients who require the use of equipment such as ventilators and blood catheters. They include *Acinetobacter*, *Pseudomonas* and various Enterobacteriaceae (including *Klebsiella*, *E. coli*, *Serratia*, and *Proteus*). They can cause serious and often fatal infections such as bloodstream infections and pneumonia [3].

Severe acquired brain injury (SABI) covers a variety of forms of central nervous system damage due to traumatic or non-traumatic causes. It results in comas of variable durations and typically causes a potentially wide range of impairments affecting physical, neurocognitive, and/or psychological functioning. It is also a significant cause of lifelong disability [4].

Healthcare-associated infections (HAIs) are medical complications and life-threatening clinical conditions that are most common after SABI in the intensive, acute and post-acute treatment phases. They should be diagnosed early as they have a negative impact on the rehabilitation process and prolong the acute phase of treatment [5].

In the intensive care setting, SABI patients may be affected by a nosocomial infection in 30% of cases and are at higher risk than other patients [6]. Prolonged hospital stay or the use of invasive devices and maneuvers may increase the incidence of infection after severe brain injury [7]. An excessive anti-inflammatory response, thought to be an adaptive and protective mechanism after cerebral ischemia, may be a facilitating factor [8]. SABI patients are more likely to be colonized by multidrug-resistant organisms (MDROs) [9].

The neurorehabilitation setting is also known to be characterized by a higher prevalence of MDRO; patients undergoing the most invasive treatments are more likely to be colonized, increasing the risk of prolonged hospital stays, exposure to antibiotic therapies and morbidity [10]. Rehabilitation units represent a challenge for infection control, also due to the dependency of patients and the environment that encourages socialization [11].

The aim of this study is to investigate and to accurately document the incidence of MDRO colonization and infection in SABI and their impact on rehabilitation outcomes. A secondary aim is to analyze the correlations of SABI’s MDRO colonization with complications, clinical parameters, and invasive medical devices in order to identify common risk factors in the SABI rehabilitation setting.

## 2. Materials and Methods

### 2.1. Study Design

This study was a retrospective observational cohort study performed in a tertiary referral specialized rehabilitation hospital in Italy. The study was approved by the regional Ethical Committee and did not receive funding.

Participants included 48 SABI patients—aged 19 to 78—consecutively admitted from January to December 2018 to a rehabilitation institute with an intensive care unit and an early rehabilitation unit. Inclusion criteria were as follows: SABI, age ≥ 18, any etiological origin, Glasgow Coma Scale (GCS) ≤ 8 for at least 24 h, and a wide range of impairments affecting physical, neurocognitive, and/or psychological functioning that involve a severe disability. No exclusion criterion was applied.

### 2.2. Data Sources and Measurements

We reviewed medical records for the presence of carbapenemase-producing Enterobacteriaceae (CPE) colonization and the development of HAI. We used a standard form to collect data. We recorded aetiology (traumatic, vascular, other), inpatient rehabilitation length of stay (LOS), presence of any co-morbid conditions and complications during inpatient rehabilitation, type and timing of MDRO infections, number of CPE transmissions, procedures performed, and invasive devices and their relation to infection. The term other etiologies refers to anoxic patients and those with metabolic failure, neoplastic damage and infective–inflammatory injury. We also recorded patients’ transfers between wards and the different types of rooms where they stayed. Two solutions were available: a room with one or two beds and an open space. The latter was characterized by 4 or 8 beds, which were close to each other without physical separation.

The cognitive and functional assessment measures—Level of Cognitive Functioning [12,13] (LCF) and Disability Rating Scale [14] (DRS)—were recorded upon admission and at discharge. The LCF Scale is one of the earlier developed scales used to assess cognitive functioning in post-coma patients. It was designed for use in the planning of treatment, tracking of recovery, and classification of outcome levels. The scale generates a classification of patients’ cognitive levels from 1 (non-responder) to 8 (purposeful–appropriate). The DRS tracks an individual from a coma to community. The measurement is possible for impairments, activity limitations, and participation restrictions. The DRS score ranges from 0 to 30. Higher values denote higher disability levels.

We used the definition of infection according to the European Centre for Disease Prevention and Control Codebook for the EU-wide Point Prevalence Survey of HAI [15], encompassing specific signs, symptoms and, if possible, microbiological confirmation. In some cases, we had no microbiological confirmation. The surveillance culture for ruling out CPE colonization was performed on admission in all patients and consisted of rectal swab culture.

### 2.3. Statistical Methods

Continuous variables were summarized using mean and standard deviation when normally distributed and using median and interquartile range (IQR) otherwise; frequencies and proportions were used for categorical and dichotomous variables. The frequency of infection was compared between groups using Fisher’s exact test and the total number of infections compared using Kruskal–Wallis test. The incidence per 1000 person days was calculated in the overall sample and in subgroups with different etiology of brain injury, using the length of hospital stay as the exposure time. Simple and age-adjusted Poisson regression models were used to identify risk factors for infection. Lastly, the association between the number of infections and the rehabilitation outcomes (DRS and LCF scores at discharge) was analyzed using linear regression models. The regression models were adjusted for baseline score on admission and for age. Analyses were conducted using Stata statistical software version 13 (Stata Corp, College Station, TX, USA), and the level of statistical significance was set at 0.05.

## 3. Results

The study sample included 48 patients aged between 19 and 78 years (mean ± SD 52.4 ± 16.2 years), mostly male (62.5%) with traumatic (37.5%) or hemorrhagic (37.5%) etiology. Some 31% of patients carried CPE on admission, while 33% were colonized during the hospitalization (Table 1).

### 3.1. HAI Incidence

A total of 101 HAI were identified in 40 patients, with an overall incidence of 10.5/1000 patient days and no significant difference among etiologies (*p* = 0.269). In some cases, pneumonia and bloodstream infections recurred twice in the same patient. In many cases, infections were device-related; 83% of patients with pneumonia had tracheotomy tube or invasive ventilation, and 67% of patients with bloodstream infection had a central venous catheter. In five of eight patients with systematic soft tissue infections, the infection source was cranioplasty.

The most frequent bacteria were *K. pneumoniae*, *P. aeuruginosa*, *A. baumannii*, *Methicillin-Resistant Staphylococcus aureus and Methicillin-Resistant Staphylococcus epidermidis*. Thirty-seven percent of patients had at least one MDRO infection, and we found a MDRO infection incidence rate of 2.8/1000 patient days. MDRO infections had a significantly (*p* = 0.003) higher incidence of 6.6/1000 patient days. The most common infections were bloodstream infections (*p* = 0.001) (Figure 1).

### 3.2. HAI Incidence

The number of HAIs was significantly correlated with the length of stay (r = 0.453, *p* = 0.001).

We found that age, previous morbidities, in-hospital colonization, and the presence of devices (indwelling catheter over 60 days, central venous catheter over 1 month, tracheotomy tube over 4 months, and PEG despite time of placement) were associated with HAI in univariate regression models. However, none of the clinical variables remained significant after adjustment for age (Appendix A).

The variables associated with MDRO HAI in univariate analyses were age, non-traumatic etiology and colonization. However, neither etiology nor colonization were significant after adjustment for age (Appendix A).

Some 97% of CPE carriers compared to 59% of non-CPE carriers developed at least one infection during hospitalization (*p* = 0.002). However, after adjusting for the length of stay, the relative risk of developing at least one infection was no longer statistically significant (RR = 1.22, 95%CI: 0.58–2.81).

In the subgroup of patients without CPE upon admission, we found a significant correlation between colonization and type of room (*p* = 0.013). Specifically, all patients located in an open-space ward were colonized during their stay, while the percentage of colonization decreased to 46% in patients who moved from open space to a double room and to 29% in patients located in a double room throughout their hospital stay.

The mean LCF was 3.9 ± 1.7 upon admission and 5.4 ± 2.0 at discharge. During rehabilitation, only two patients worsened, 19% were stable and 77% improved. We found a negative association between the number of HAI and LCF change from admission, so when HAI increased, the change in LCF from baseline was smaller (b = −0.25; 95%CI: −0.50–−0.01). The association was not significant after adjustment for age. Similar results were found for MDRO HAI.

The mean DRS score was 18.7 ± 6.2 upon admission and 13.4 ± 7.6 at discharge. Similar to what was found for LCF score, only two patients worsened; 13% were stable and 80% improved. DRS at discharge increased with HAI number (b = 0.92; 95%CI: −0.01–1.84); however, the association was not significant after adjustment for age. Similar results were found for MDRO HAI.

Complications and HAI significantly affected length of stay. The results indicate that patients with MDRO HAI and those with complications had a similar length of stay (Figure 2).

## 4. Discussion

In this study, we described CPE colonization and MDRO infections in an Italian neurorehabilitation hospital over 1 year in a cohort of SABI patients with heterogeneous etiology, severe disability and total dependency, and cognitive and behavioral deficits. Our study suggests that CPE carriers may be at risk of HAI and worse outcomes compared with non-CPE carriers. 

These data were included in a multicenter study [16] comparing similar SABI rehabilitation units linked by a common health and clinical pathway. These units are part of a clinical care pathway for SABI patients that covers the acute and rehabilitation phases up to discharge to home or community settings. All these units are subject to similar prevention and infection control measures in accordance with the clinical care pathway. The two semi-intensive care units interact with the intensive care unit and neurosurgery to ensure a timely neurorehabilitation approach. They define diagnosis, begin the rehabilitation process, and, once they are stable, transfer patients to the next level of care. The two post-acute units provide comprehensive care for SABI patients, offering separate units for disorders of consciousness with a long-term rehabilitation process. Patients who are not suitable for intensive rehabilitation are transferred to the long-term care unit. All five wards use the same diagnostic and treatment approach, with an individualized rehabilitation plan that involves the patient’s family. 

This study included 134 SABI patients treated in inpatient rehabilitation settings, such as semi-intensive care units, post-acute care units, and long-term care facilities. The aim of the study was to estimate the incidence of HAIs in different SABI rehabilitation settings and to determine the risk factors and impact of HAIs on neuromotor and cognitive recovery. The incidence of HAIs and MDRO HAIs was significantly higher in the semi-intensive care unit than in other settings. Risk variables for HAIs and MDRO HAIs included older age, more devices, and CPE colonization, while a high plasma prealbumin level appeared to have a protective effect. This study confirmed that nosocomial infections are associated with increased length of stay and that colonization is associated with poor prognosis and functional outcomes, with reduced ability to achieve cognitive self-care, employability, and independent living. The need to ensure the protection of non-colonized patients, particularly those with severe disabilities upon admission, was therefore highlighted herein.

A previous Italian multicenter study [17] included 228 SABI patients and compared infected and uninfected patients. In this study, the presence of multiple bacterial species in the same culture specimens was most common (58.1%), and 55.5% of patients required functional isolation due to multidrug-resistant bacteria. The functional status of both groups improved after rehabilitation, but multivariable analysis showed that the infected group of patients improved significantly less than the uninfected group. Length of stay and number of missed rehabilitation sessions were not statistically different between the groups; mortality was significantly higher in the infected group. Infected patients with SABI had a longer length of stay, significantly higher mortality, and worse functional outcome than uninfected patients. The most common CPE was *K. pneumoniae*, producing KPC-type carbapenemase (KPC-KP) as in the rest of the country [18]. 

The latter article reported the results of a nationwide cross-sectional survey conducted from 15 May to 30 June 2011 to investigate the prevalence of carbapenemase-resistant Enterobacteriaceae in Italy and to characterize the most common resistance mechanisms and their distribution patterns. In Italy, a rapid and significant increase in carbapenem-susceptible K. pneumoniae CRE was reported by most of the participating laboratories (23 out of 25).

Tedeschi et al. reported the high burden of CRE colonization and infection in SABI and spinal cord injury patients over a 6-year period (2012–2017) [19]. 

During the study period, the authors observed 4180 patients with a mean length of stay of 79 ± 4 days, for a total of 333,484 patient days. Overall, 9.3% of patients were carriers of rectal CPE on admission, and 8.1% acquired colonization during hospitalization. CPE-BSI was diagnosed in 96 of the 699 colonized patients (14%). No CPE-BSI occurred in non-colonized patients, and no other CPE infections were identified during the study period. Trends in CPE prevalence on admission, CPE colonization, and CPE-BSI incidence were discontinuous and varied over the 6-year observation period.

The authors argued that the application of infection control measures was effective in reducing in-hospital CPE transmission when the number of imported CPE carriers was lower. The study highlighted the difficulties in maintaining the results over time, especially given the increase in colonized patients entering the hospital and the difficulties in routinely applying infection control measures such as strict patient isolation. In this setting, the potential solution of spatial isolation is difficult to achieve as the main goal of the rehabilitation program is reintegration into the community and participation in social activities to the extent that the acquired disability allows.

Regarding the burden of CPE colonization, mainly KPC, we found an increased risk of infection in colonized compared to non-colonized patients. They were therefore at risk of prolonged hospitalization. Similarly, in their 2017 study, Zembower et al. concluded that patients with brain and spinal cord injuries harbor a significant percentage of the CPE population, particularly *K. pneumoniae* producing KPC-type carbapenemase in the United States. Due to repeated hospitalizations and prolonged colonization, they represent a significant reservoir of these multi-drug-resistant bacteria [20]. This study analyzed data from 218 patients with isolated KPC-KP over a 6-year period (2009–2014), of whom 86 (39%) were brain and spinal cord injury patients. Of these 86 patients, 27 (31%) had more than one isolated KPC-KP over time. These patients often had multiple readmissions. The average number of readmissions in the year before KPC-KP isolation was 2, while the highest number was 12. Given the frequency of readmissions, these patients represent a repeated source of potential transmission within the healthcare setting. Selective screening of high-risk patient populations such as these may reduce the risk of KPC-KP transmission. Because of the high morbidity and mortality associated with KPC-KP infections, patients are often treated with broad-spectrum antimicrobials or combinations of antimicrobials. However, it is important to distinguish between colonization and KPC-KP infection, as these agents often have significant side effects, and treating colonization can lead to further antibiotic resistance without any therapeutic benefit.

While the effect of colonization on rehabilitation outcome was not considered in the cited study [17], the same author [21] wrote about colonization in 2020 and concluded that HAIs and microbial colonization were common in patients with SABI admitted to neurorehabilitation units. Patients with colonized SABI had a similar functional outcome to those without infection. However, they had a higher mortality rate and a significantly longer LOS than those without infection. We hope that future studies will aim to investigate how rehabilitation environments should be organized to prevent the spread of colonization and HAIs.

The change in LCF and DRS score between admission and discharge seems to be related to the development of HAI but not to complications. However, we found that age was a confounding factor associated with worse outcomes. Infections can adversely affect outcomes, including increased mortality or prolongation of rehabilitation LOS [22]. We found that LOS was significantly increased in the presence of complications and HAI. The data suggest that LOS in the presence of MDRO HAI was similar to that in patients with complications. However, we only recorded one death among all participants, and this was not infection-related despite the severity of the patient’s condition. In our study, 83% of patients developed at least one HAI during hospitalization, and 37% developed an MDRO HAI. In total, we found 101 HAIs and 27 MDRO HAIs. The HAI incidence was 10.5/1000 patient days, and the MDRO HAI incidence was 2.8. The incidence was lower in trauma patients and higher in anoxic and metabolically ill patients, who had 6.6 HAIs/1000 patient days.

Pneumonia and bloodstream infections affected 37% of patients and were mostly device-related (83% pneumonia and 67% bloodstream infections), in line with previous studies [23,24]. Pneumonia and bloodstream infections recurred at least twice in the same patient, and MDRO bloodstream infections were more common in patients with etiologies other than traumatic and hemorrhagic. Age, previous comorbidity, colonization, prolonged presence of an indwelling catheter for more than 60 days, a CVC for more than 30 days, a tracheostomy tube for more than 120 days, and PEG regardless of time were potential risk factors for HAI. 

Systemic soft tissue infections occurred in 17% of patients and were associated with cranioplasty in 60% of cases. They are a well-documented complication of cranioplasty, occurring in 10.3–26.4% of cases [25]. Cranioplasty was thought to have a protective and cosmetic role, but recent studies have shown that it also improves neurological outcomes [26]. However, it is associated with several potential complications, such as infection, which can lead to a prolonged hospital stay and poor outcomes [27]. Despite the predictable risks, our patients required neurosurgical treatment to achieve their desired outcome. These findings have encouraged us to pay more attention to infection control measures in the postoperative period and to increase neurosurgeons’ awareness of this issue.

In 2015, Hayden et al. described the implementation of a bundled intervention to reduce KPC colonization and infection in long-term acute care hospitals, which resulted in a sustained reduction in KPC cross-transmission and blood infections [28]. In this study, the authors used a stepped design to test whether a combined intervention (screening of patients for rectal KPC colonization on admission and every two weeks; contact isolation and geographical segregation of KPC-positive patients in ward cohorts or single rooms; daily bathing of all patients with chlorhexidine gluconate; and education of staff and monitoring of adherence) would have reduced KPC colonization and infection in four long-term acute care hospitals with a high endemic prevalence of KPC.

The prevalence of KPC colonization decreased early during the intervention and then plateaued. During the intervention, while the prevalence of KPC admission remained high, the incidence rate of KPC colonization decreased from four to two acquisitions per 100 patient weeks during the intervention, as did the presence of KPC in any culture specimen. A bundled intervention was associated with clinically important and statistically significant reductions in KPC colonization, KPC infection, bacteremia of any cause, and blood culture contamination in a high-risk long-term acute care hospital population.

The authors suggested that a focus on infection control measures is critical, especially for SABI patients; otherwise, rehabilitation efforts will be frustrated by cross-transmission of CPE or MDRO infection, which we have shown to have an important impact on recovery. During inpatient rehabilitation, we need to protect the frail patient as an SABI patient, and we should make every effort to prevent cross-transmission of CPE and infections. We found the number of HAIs to be associated with poor cognitive and disability outcomes. However, these associations were confounded by age and were no longer significant after age adjustment. In addition, we cannot say that HAIs increase length of stay, as the opposite may be true. In a recent paper, Bellaviti et al. confirmed that the risk of nosocomial infections in high-acuity rehabilitation units cannot be reduced to zero and may increase with increasing patient complexity. The authors concluded that the impact of systemic infections due to Gram-negative bacteria significantly worsens discharge outcomes [29]. This study was conducted in an Italian intensive rehabilitation unit with the aim of investigating the influence of systemic infection, particularly with regard to multidrug-resistant bacteria, and analyzing the impact of comorbidities as a risk factor for the development of systemic infection on rehabilitation outcomes in patients with severe brain injury. This research is a prospective observational cohort study comparing 221 SABI patients with and without systemic infection (at least one positive blood culture) in terms of rehabilitation outcomes. Length of hospital stay and the role of comorbidities were also taken into account. The group of patients with systemic infections (mainly due to Gram-negative bacteria) had significantly worse outcomes and hospital stays more than two times longer. However, patients with at least one positive blood culture were found to have a lower functional status on admission. The authors concluded that more attention should be paid to the impact of infection during inpatient rehabilitation, with specific procedures and appropriate environments for preventing the spread of infection.

## 5. Conclusions

The main limitations of our study are that it is monocentric and retrospective. This may affect the completeness of the information available. In addition, the small sample size may limit our ability to detect significant associations. 

Further prospective studies based on larger samples are needed to investigate in deeper detail the association between infection control interventions and CPE cross-transmission and MDRO infections and their impact on outcomes in SABI rehabilitation hospitals.

## Figures and Tables

**Figure 1 microorganisms-12-00830-f001:**
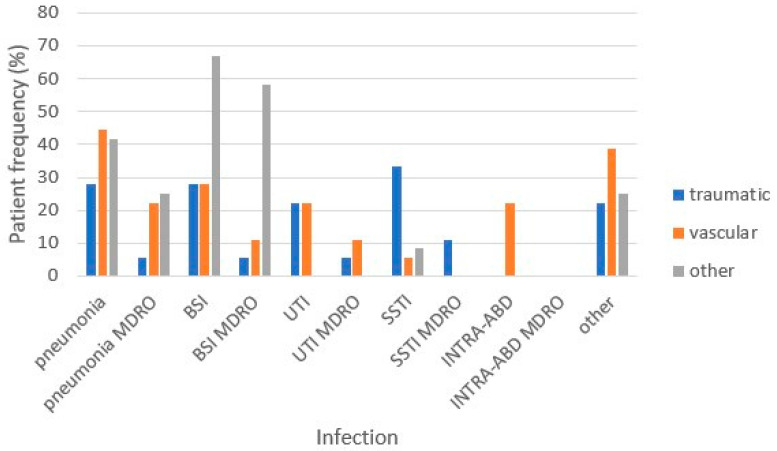
Rate of pts/HAI-MDRO HAI type and etiology of severe acquired brain injury. MDRO: multidrug-resistant organisms, BSI: bloodstream infection, UTI: urinary tract infection, SSTI: systemic soft tissue infection, INTRA-ABD: intra-abdominal infection.

**Figure 2 microorganisms-12-00830-f002:**
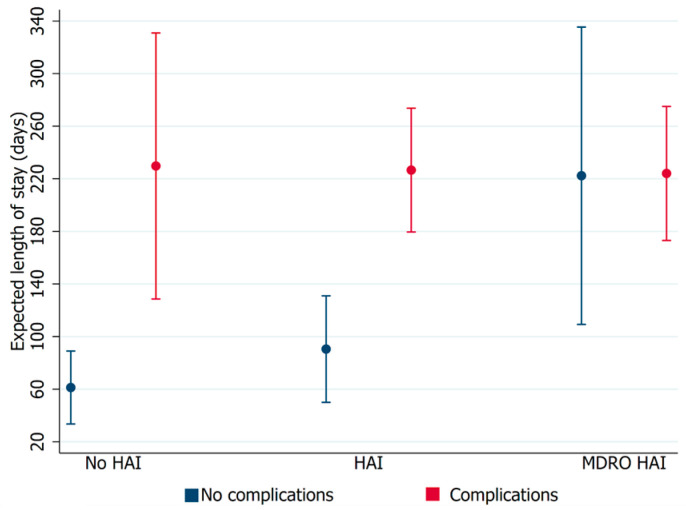
Relationship between length of stay and the presence of complications in patients with healthcare-associated infections (HAIs) or multidrug-resistant organism healthcare-associated infections (MDRO HAIs). Error bars represent 95% confidence intervals.

**Table 1 microorganisms-12-00830-t001:** Sample characteristics.

Study Sample	*N* = 48
**Male, *n* (%)**	30 (62.5%)
**Age, mean ± SD**	52.4 ± 16.2
**Etiology, *n* (%)**	
Traumatic	18 (37.5%)
Associated trauma *	16 (88.9%)
Vascular	18 (37.5%)
Other (anoxic, metabolic, neoplastic and infective–inflammatory)	12 (25%)
**Comorbidity, *n* (%)**	10 (20.8%)
**Pre-admission surgery**	
No	19 (39.6%)
Neurosurgery	26 (54.2%)
Other (abdominal and skeletal surgery)	3 (6.2%)
**LCF upon admission, mean ± SD**	3.9 ± 1.7
**DRS upon admission, mean ± SD**	18.7 ± 6.2
**Transfers, *n* (%)**	23 (47.9%)
**Surgery during hospitalization**	
**No**	26 (54.2%)
**Neurosurgery**	15 (31.2%)
**Other (abdominal and skeletal surgery)**	7 (14.6%)
**CPE colonization, *n* (%)**	
**No**	17 (35.4%)
**Upon admission**	15 (31.3%)
**During hospitalization**	16 (33.3%)
**Complications, *n* (%)**	37 (77.1%)
**Length of stay, median (IQ range)**	206 (122–273)
**Medical devices**	
**Indwelling catheter, *n* (%)**	45 (93.8%)
**days, median (IQ range)**	71 (24–107)
**Central venous catheter, *n* (%)**	27 (56.3%)
**days, median (IQ range)**	41 (7–58)
**Tracheotomy tube, *n* (%)**	41 (85.4%)
**days, median (IQ range)**	135 (50–233)
**PEG or PEJ, *n* (%)**	30 (62.5%)
**days, median (IQ range)**	234 (122–299)

* Among patients with traumatic etiology. LCF: levels of cognitive functioning, DRS: disability rating scale, CPE: carbapenemase-producing Enterobacteriaceae, PEG: percutaneous endoscopic gastrostomy, PEJ: percutaneous endoscopic jejunostomy.

## Data Availability

The data presented in this study are available on request from the corresponding author due to privacy reasons.

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
