# Peer review of "Impact of Multidrug-Resistant Organisms on Severe Acquired Brain Injury Rehabilitation: An Observational Study"

_microorganisms, 2024, doi:10.3390/microorganisms12040830_

Round 1

Reviewer 1 Report

Comments and Suggestions for Authors

I have read with great interest the manuscript titled “Safe & Sound. Which Rehabilitation with Infections? Impact of Multi-Drug Resistant Organisms on Severe Acquired Brain Injury Rehabilitation, an Observational Study”, and I believe it is generally suitable for publication in Microorganisms, Special Issue “Antimicrobial Resistance: Challenges and Innovative Solutions”. The study offers valuable insights into the prevalence and impact of MDRO colonization and HAI in SABI rehabilitation. However, similar studies have been conducted previously, which slightly diminishes the novelty of the research. Given the increasing threat of antimicrobial resistance and the vulnerability of SABI patients to infections, the topic addressed in the article holds significant importance. Understanding the incidence and risk factors associated with MDRO colonization and HAI in this population is crucial for improving patient outcomes and guiding infection control measures. The manuscript is well-organized and clearly presents the methodology, results, and conclusions. The use of appropriate statistical analyses enhances the credibility of the findings. However, the small sample size limits the generalizability of the findings. The article is likely to be of interest to researchers and healthcare professionals involved in infection control and rehabilitation medicine. The findings highlight the importance of implementing effective strategies to prevent and manage MDRO colonization and HAI in SABI rehabilitation settings. However, I have a few minor comments and suggestions.

Comments:

1) I think the title of the article, which is broken into three sentences, can be rephrased to make it clearer and better reflect the aim of the study.

2) Lines 43,44. "They can cause severe and often deadly infections such as bloodstream infections and pneumonia [3]". The internet page does not open through link #3 in References. Please check or replace link #3 in References.

3) Table 1. "PEG or PEJ, n (%)." Please expand these abbreviations in the manuscript.

4) Figure 1. All abbreviations used in the figure should be explained in the caption.

5) Lines 184, 269. “MRDO”. Please correct to MDRO.

6) Figure 2. In the caption, it should be clarified what the error bars represent - standard deviations or standard errors.

Author Response

1) I think the title of the article, which is broken into three sentences, can be rephrased to make it clearer and better reflect the aim of the study.

We acknowledged the suggestion to rephrase the title, that now reads: Impact of Multidrug Resistant Organisms on Severe Acquired Brain Injury Rehabilitation, an Observational Study.

2) Lines 43,44. "They can cause severe and often deadly infections such as bloodstream infections and pneumonia [3]". The internet page does not open through link #3 in References. Please check or replace link #3 in References.

We tried the link and found that it works, so we reported this problem to the Assistant Editor.

3) Table 1. "PEG or PEJ, n (%)." Please expand these abbreviations in the manuscript.

We have now expanded the abbreviations in Table 1, as suggested.

4) Figure 1. All abbreviations used in the figure should be explained in the caption.

We have now explained in the caption the abbreviations used in the figure

5) Lines 184, 269. “MRDO”. Please correct to MDRO.

We apologize for the typo, that has now been corrected

6) Figure 2. In the caption, it should be clarified what the error bars represent - standard deviations or standard errors

The error bars represent 95% confidence interval of the estimated length of stay.

Reviewer 2 Report

Comments and Suggestions for Authors

The presented study is an interesting piecec of research, based on statistical analysis of past data. There are a few issues that I suggest need to be resolved, but apart from these mentioned below - I rate this manuscript high.

The name is multidrug resistant, not multi drugs resistant. I also suggest to use the term bacteria rather than organisms, as the Authors only refer to these microorganisms in their study.

Abstract

l. 23: found, not founded

Introduction

Although there are some language flaws, the Introduction reads well. It is comprehensively written, all necessary information provided.

The aim is precisely specified.

Materials and Methods

I suggest to divide this section into subsections that would provide a more readable structure of this section

Results

l. 137-139: the Latin names of bacteria should be corrected: written in italics and the species name should start with small letter.

Figure 1 – the abbreviations used in the figure should be explained.

Both tables that show detailed results of statistical analysis could be put as supplementary data, while in the main body of the text you could leave only the results with the highest statistical significance. This would make the results presented in this way much more comprehensible.

Conclusions

I suggest to restructure this section. In the first fragment place the conclusions that were drawn from the study, then the reference to other Authors’ results, then limitations of this study and finally the need for further studies.

Comments on the Quality of English Language

The English language used in the manuscrip is quite good, but the text definitely needs refining.

Author Response

The name is multidrug resistant, not multi drugs resistant.

Thank you for noticing the typo, that has now been corrected

I also suggest to use the term bacteria rather than organisms, as the Authors only refer to these microorganisms in their study.

We have retained the abbreviation and used the term bacteria in the text.  

Abstract

  1. 23: found, not founded

We have corrected the typo, thanks

Introduction

Although there are some language flaws, the Introduction reads well. It is comprehensively written, all necessary information provided.

The language has been revised throughout the manuscript

The aim is precisely specified.

Materials and Methods

I suggest to divide this section into subsections that would provide a more readable structure of this section

We acknowledged the suggestion and added the follow subtitles: study design, data sources and measurements, statistical methods.

Results

  1. 137-139: the Latin names of bacteria should be corrected: written in italics and the species name should start with small letter.

We checked and we corrected the Latin names according to your suggestion, thank you.

Figure 1 – the abbreviations used in the figure should be explained.

We have now explained the abbreviations used in the figure

Both tables that show detailed results of statistical analysis could be put as supplementary data, while in the main body of the text you could leave only the results with the highest statistical significance. This would make the results presented in this way much more comprehensible.

We agree and moved tables 2 and 3 to the supplementary materials.

Conclusions

I suggest to restructure this section. In the first fragment place the conclusions that were drawn from the study, then the reference to other Authors’ results, then limitations of this study and finally the need for further studies.

We are grateful to the reviewer for this comment, that gave us the opportunity to restructure the discussion

Comments on the Quality of English Language

The English language used in the manuscript is quite good, but the text definitely needs refining.

We have revised the English language

Round 2

Reviewer 1 Report

Comments and Suggestions for Authors

The authors have taken into account the comments and significantly improved the manuscript. I believe that the manuscript can now be published in its present form.

Reviewer 2 Report

Comments and Suggestions for Authors

Dear Authors,

Thank you for correcting your manuscript.

All is fine.